# Development and Characterization of Membranes with PVA Containing Silver Particles: A Study of the Addition and Stability

**DOI:** 10.3390/polym12091937

**Published:** 2020-08-27

**Authors:** Audie K. Thompson, Cannon Hackett, Tony L. Grady, Silver Enyinnia, Quincy C. Moore, Felecia M. Nave

**Affiliations:** 1Department of Chemical Engineering, University of Arkansas, Fayetteville, AR 72701, USA; cjhacket@uark.edu; 2Department of Chemistry, Prairie View A&M University, Prairie View, TX 77446, USA; tlgrady@pvamu.edu; 3Department of Chemical Engineering, Prairie View A&M University, Prairie View, TX 77446, USA; silver.enyinnia@gmail.com; 4Department of Biology, Prairie View A&M University, Prairie View, TX 77446, USA; qcmoore@pvamu.edu; 5Department of Chemistry, Alcorn State University, Lorman, MS 39096, USA; fmnave@alcorn.edu

**Keywords:** ultrafiltration, biofouling, poly(vinyl) alcohol, silver particles

## Abstract

Developing technologies for the reduction of biofouling and enhancement of membrane functionality and durability are challenging but critical for the advancement of water purification processes. Silver (Ag) is often used in the process of purification due to its anti-fouling properties; however, the leaching of this metal from a filtration membrane significantly reduces its effectiveness. Our study was designed to integrate the positive characteristics of poly vinyl alcohol (PVA) with the controlled incorporation of nano-scale silver ions across the membrane. This approach was designed with three goals in mind: (1) to improve antifouling activity; (2) to prevent leaching of the metal; and (3) to extend the durability of the functionalized membrane. The fabrication method we used was a modified version of manual coating in combination with sufficient pressure to ensure impregnation and proper blending of PVA with cellulose acetate. We then used the spin coater to enhance the cross-linking reaction, which improved membrane durability. Our results indicate that PVA acts as a reducing agent of Ag^+^ to Ag^0^ using X-ray photoelectron spectroscopy analysis and demonstrate that the metal retention was increased by more than 90% using PVA in combination with ultraviolet-photo-irradiated Ag+ reduced to Ag^0^. The Ag^+^ ions have sp hybrid orbitals, which accept lone pairs of electrons from a hydroxyl oxygen atom, and the covalent binding of silver to the hydroxyl groups of PVA enhanced retention. In fact, membranes with reduced Ag displayed a more effective attachment of Ag and a more efficient eradication of *E. coli* growth. Compared to pristine membranes, bovine serum albumin (BSA) flux increased by 8% after the initial addition of Ag and by 17% following ultraviolet irradiation and reduction of Ag, whereas BSA rejection increased by 10% and 11%, respectively. The implementation of this hybrid method for modifying commercial membranes could lead to significant savings due to increased metal retention and membrane effectiveness. These enhancements would ultimately increase the membrane’s longevity and reduce the cost/benefit ratio.

## 1. Introduction

The search for more rapid and effective water filtration membranes remains a vital area for research. Biofouling, especially microbial accumulation, is detrimental to water filtration processes and decreases the durability of membranes. Both organic and biofouling are attributed to polysaccharides [1]. Therefore, membranes that are resistant to the collection of microorganisms are essential for water purification. Copper has been used to prevent microorganism accumulation [2], and magnesium affects polysaccharide fouling [1,3]. Microbes bind strongly to metals via sulfhydryl groups [4], thus leading to pollutant degradation. It is well known that hydrophilic surface modification can be an effective method for reducing membrane fouling since the hydrophilic surface can repel foulant adsorption via the repulsive hydration force [5]. In addition, silver is widely known as a biocide and is used in numerous applications such as air filters, medicinal materials, drug delivery, textiles, water filtration, food packaging, and wastewater treatment [6,7,8,9]. Besides enhancing the membrane’s antimicrobial properties, silver is known to affect a membrane’s hydrophilicity as well. Andrade et al. embedded silver nanoparticles (AgNPs) into polysulfone nanofiltration membranes via the in-situ method and ex-situ method and found that the addition of silver onto the membrane’s surface reduced the contact angle of the membrane from 75.7° to 60.9° and increased the pure water flux of membranes from 25 to 121 L/m^2^h [10]. Optimization of the capture and stabilization of metals such as silver, which exhibits exceptional antimicrobial activity [7,8,11,12], has been a key focus for recent studies on filter substrate structure. For example, preventing Ag^+^ from leaching into the permeate and causing loss of antimicrobial properties [13] is critical to silver’s effectiveness.

Studies focused on the reduction of silver composites using chemical or ultraviolet (UV) irradiation are promising, and synthetic routes for embedding silver nanoparticles in various polymers have been implemented [14,15] to combat leaching. Polymers are exceptional host materials for metals, and when presented as polymeric solutions, their capacity for capturing and enabling metal ions to become semi-stable in the matrix is particularly useful. Random attachment of nanoparticles on the polymer surface following exposure to a silver nanoparticle solution embeds these particles in the polymer [16,17]. This process allows the polymer to serve as a surface capping agent that prevents agglomeration and precipitation of the particles [14]. A recent study successfully used hybrid thin-film composite membranes with embedded catalytic silver nanoparticles to effectively filter contaminants while enhancing the antimicrobial activity of the membrane [18].

One of the polymer solutions used to embed AgNPs is aqueous poly (vinyl alcohol) (PVA), which when exposed to a silver salt solution under reducing conditions allows random surface attachment and embedding of the silver ion nanoparticles in the polymer [14]. Although PVA hydrogels provide smooth hydrophilic surfaces with limited protein adsorption [19,20], the water-soluble PVA molecules and the metal cations interact in aqueous solution. This interaction challenges the stability of the composite due to weak ion binding on the surface and complete ion loss after a few washes [21]. By itself, the PVA polymer cannot meet the needs for optimal water filtration, despite studies showing that UV irradiation lowers the oxidation state of Ag^+^ to metallic Ag^0^ [22] and that UV photon absorption promotes silver nanoparticle (AgNP) growth on cellulose fibers by a mechanism that involves breakage of the oxygen bridge between the monomers and the formation of aldehyde groups that reduce Ag^+^ ions [23,24].

Another polymer that is widely used for many fabrication processes, especially for water purification, is cellulose acetate (CA). Both PVA and CA have been widely used for polymer nanocomposites due to easy processability. A study investigating the properties of PVA hydrogels spin-coated onto regenerated cellulose demonstrated that these two polymers could be successfully combined for use in pure water and ultrafiltration flux. For a given thickness of the polymeric composite, increasing the degree of cross-linking had no measurable effect on flux or the extent of fouling. It was also demonstrated that the success of the surface modification between the two substrates is significantly improved by increasing the hydrophilicity of membrane surfaces using methods such as grafting, chemical cross-linking, or chemical reaction of the surface. This change reduces the adsorption of foulant by inhibiting non-specific binding between the membrane surface and retained molecules, particularly proteins [19,25]. In fact, the ideal membranes for water purification (UF or RO) are hydrophilic semi-permeable membranes that allow for high flux while being highly selective through the membrane [26]. CA’s surface has been chemically reacted and modified with numerous high-performance surface modifiers to increase its antifouling capabilities, and PVA has attracted interest due to the ease of its surface modification. In order to decrease the swelling of the polymeric membrane in water, a chemical solution such as glutaraldehyde (GA) is employed to cross-link the PVA polymer [20,27]. GA is widely used in cross-linking PVA since the aldehyde group of GA forms acetal linkages with the hydroxyl groups of PVA and enhances the potential for covalently linking the PVA to the CA [19,20,28]. Adjusting the cross-linking ratio also controls porosity and swelling, which can be used to control flux [27].

This paper focuses on the use of nanotechnology, specifically the use of AgNP, for water purification purposes. It describes the properties and methods used for growing the AgNP on PVA filters and addresses the physicochemical properties such as Ag binding and reduction of Ag+ to Ag^0^ that contribute to the strength of attachment, decreased leaching, and prevention of microbial growth. In this study, membranes were fabricated using a novel hybrid method that consisted of coating 8% (*w*/*v*) PVA cross-linked with 2.5% (*w*/*v*) GA solution on a commercially purchased CA membrane with 45μm symmetrical pores. Crosslinking PVA with GA forms a mesh-size matrix of pores in the range of ultrafiltration. Hybrid membranes were functionalized using a tridentate metal chelator, iminodiacetic acid (IDA), and a fouling experiment was performed to test the rejection of the protein, bovine serum albumin (BSA). The presence of silver was characterized using X-ray photoelectron spectroscopy (XPS) and atomic absorption spectroscopy (AAS). Using UV irradiation to impregnate the membranes with silver enabled the synthetic polymeric membrane to be more hydrophilic. The impact of utilizing PVA to stabilize silver was evaluated for bond formation, fouling, and flux performance. The novelty of our approach is that UV irradiation promotes PVA’s capacity for reducing and thus stabilizing silver.

## 2. Materials and Methods

Polyvinyl alcohol (Sigma Aldrich Chemical Company, St. Louis, MO, USA) was purchased as a fully hydrolyzed powder with an average molecular weight of 85,000 g/mol. For the hybrid method, CA membranes were commercially purchased from Fisher Scientific Company (Waltham, MA, USA) with a symmetrical pore size of 0.45 µm and a diameter of 47mm. The tridentate chelator, iminodiacetic acid (IDA), and 50% (*w*/*w*) GA were also purchased from Sigma-Aldrich Chemical Company. Bovine serum albumin, sulfuric acid, copper sulfate, silver nitrate, ethylenediaminetetraacetic acid (EDTA), acetic acid, and sodium bicarbonate were purchased from Fisher Scientific Company. All materials were used without further purification.

### 2.1. Membrane Fabrication—Hybrid Method

An 8% (*w*/*v*) PVA casting solution was prepared by mixing 750 µL of the PVA solution with 170.4 µL of 10% (*v*/*v*) sulfuric acid, 27.9 µL of 10% (*v*/*v*) acetic acid, 27.9 µL of 50% (*v*/*v*) methanol, and 85.2 µL of 2.5% or 5% (*v*/*v*) GA. Three separate casting methods—Manual, Dynamic, and Spin Coater—were combined to create the hybrid coating method. This hybrid method was developed to optimize the asymmetry of the membrane by impregnation and proper blending of PVA with CA. The commercially purchased CA support membrane was saturated with deionized water and placed on a glass plate for casting. PVA/GA casting solution was then poured and spread across the support membrane’s surface. For the PVA/GA reaction, the membranes were allowed to sit for approximately 8 and 6 min for 2.5% GA and 5% GA, respectively. Membranes will be denoted as PVA-2.5 and PVA-5 for GA percentage used. The membrane casting time was determined by allowing the casting solution to become more viscous to form a hydrogel. However, the membranes were not allowed to completely dry for the subsequent steps in the membrane fabrication process. Following the manual coating method, membranes were placed into the dead-end ultrafiltration system. Fifty psi of pressure was applied to the membranes for 30 s to ensure the dynamic impregnation of the inner pores of the support membrane with the casting solution. The last step in the hybrid method is the spin coater Laurell WS-650-23 Spin Coater, which consists of a 9-step process to minimize losing PVA and optimize membrane impregnation using centrifugal forces, thus optimizing membrane asymmetry. The RPM of the 9-step process was increased in increments of 50 from 100 RPM to 500 RPM, reducing the residence time by 30 s. The last two steps were used as drying steps to ensure complete cross-linking. Afterward, the membrane was placed in deionized water to cease the reaction, to rinse off any excess glutaraldehyde, and to allow for swelling.

IDA was attached to the membrane by placing the membrane in 17 mL of 2 M sodium bicarbonate solution and 1.25 g of IDA. The membrane and solution were then incubated in a heater shaker at 60 °C for 24 h. Afterward, the membrane was soaked in deionized water for 6 h to allow for swelling. Functionalizing the membrane with Ag^+^ occurred by placing the membrane in 40 mL of a 0.05 M silver nitrate solution for 6 h. After this time, the membrane was dipped in 400 mL of deionized water to remove any unbound silver. These membranes will be denoted as PVA-Ag.

In addition to attaching Ag^+^, membranes were photo-reduced using UV light to convert silver ions to Ag°. After removing the membrane from AgNO_3_ and rinsing off any unbound silver ions, membranes were placed on a petri-dish and inserted into a Spectroline Fluorescence Analysis Cabinet (Spectroline, Westbury, NY, USA) where a longwave UV light at 365 nm was used to irradiate the membrane for approximately one minute. Irradiated membranes will be denoted as PVA-UV.

### 2.2. Fourier Transform Infrared Spectroscopy

Data from a Fourier Transform Infrared Spectrometer (FTIR) (PerkinElmer, Waltham, MA, USA) was used to analyze the chemical structure of dried membranes. Membranes were dried overnight at room temperature for analysis. Each sample was scanned a total of 32 times to retrieve spectra frequencies, which provided peaks depicting functional groups within the PVA coating. The frequency bands acquired in the range of wavenumber 4000 to 650 cm^−1^ represent stretching, vibrations, narrowing of the compounds, and the bonds formed between molecules [29]. For the hybrid method, membranes were investigated on both sides: The top side, which was coated with PVA, and the bottom side, which was not coated with PVA.

### 2.3. Scanning Electron Microscopy and Energy Dispersive X-ray Spectroscopy

Elemental analysis of the dried membrane’s surface was performed via energy dispersive X-ray spectroscopy (EDS) and pore morphology was observed using a scanning electron microscope (SEM) by FEI, model Nova Nanolab 200 Dual-Beam Focused Ion Beam (FEI, Hillsboro, OR, USA). Membranes were dried in a petri-dish overnight and then sputter-coated with platinum before being loaded into the SEM.

### 2.4. Atomic Absorption Spectroscopy

Membrane samples were analyzed for metal retention using a Varian spectra atomic absorption spectrometer (AAS) model 200 FS (LabX, Midland, ON, Canada). For metal extraction, membranes were placed in 20 mL of a 0.05 M EDTA overnight. EDTA has a higher affinity for metals due to its six chelating ligands as compared to three ligands on IDA. This enabled the metals to be stripped from the membrane once immersed in the EDTA solution. This procedure provides an approximate quantitative determination of metal concentration as compared to an exact quantitative determination due to factors including metals reattaching back to IDA on the membrane and metals attached within the pores of the cellular matrix. Samples of the effluent were collected, and the concentration of silver were measured with AAS.

### 2.5. X-ray Photoelectron Spectroscopy

A Perkin Elmer PHI 5600ci ESCA System (Waltham, MA, USA) with a non-monochromatized Mg Kα radiation (hν = 1253.6 eV) produced with a power of 300 W was used for analyses. The analyzer pass energy was 187.85 eV for survey spectra and 58.70 eV for the multiplex spectra. A 0.6 mm analyzed area was used. The following core levels were analyzed: C1s, O1s, Ag3d, and Ag M.N.N. Data were recorded with the PHI MultiPak Version 6 1A software for XPS. The flood gun was used for charge correction. The binding energy (BE) was corrected considering the charge shift observed for the C1s peak that should be centered at 284.6 eV.

### 2.6. Equilibrium Solution Content

The Equilibrium Solution Content (ESC) examines the transport environment of water in membranes and the adsorption of water to membranes. Additionally, it provides data utilized to estimate the mesh size of the membranes [30]. First, samples of different membrane samples were soaked in conical tubes containing deionized water at room temperature for 24 h. After that, the membrane’s weight was measured after pat drying it with a Kimwipe. After measuring the wet weight, membranes were dried at 100 °C for approximately 2 h; then, the dry weight was measured. The equation for *ESC* is given as:(1)$$ESC=\left(\frac{W_{wet}-W_{dry}}{W_{wet}}\;\right)\;\times \;100\%$$
where *W**_wet_* represents the wet weight of membranes and *W_dry_* is the dry weight of membranes.

### 2.7. Permeability and Retention

A dead end-filtration system was used to measure the permeate flux. The feed was transported through the membrane by a pressure driving force. The flux experiments for membranes fabricated via the hybrid method were carried out in a Millipore model 8050 dead-end ultrafiltration cell and for membranes fabricated via phase inversion, a stainless-steel Sterlitech HP4750 Stirred Cell was used. Both were connected to a nitrogen cylinder, which provided a transmembrane pressure of 50 psi. A magnetic stirrer was used to stir the feed to decrease the effect of concentration polarization on the flux. Membrane performance were all characterized by measuring two main parameters: Pure water flux (L/m^2^·h) and protein rejection (*R*). Flux, *J*, is used to determine flux measurements.
(2)$$J=\;\frac{m}{A\;\times \;\Delta t}$$
where *m* is the mass of the permeate in a liter (L), *A* is the membrane area, and *t* represents time in an hour (h). The protein rejection was calculated using:(3)$$R=\left(1-\;\frac{C_{permeate}}{C_{feed}}\right)\;\times \;100\%$$
where *C**_permeate_* and *C**_feed_* represent the concentration of the permeate and feed solution, respectively. A BSA solution (2.0 mg/mL) was used as a feed solution. The concentration of the permeate was determined by acquiring the absorbance of the permeate at 280 nm using UV-vis.

### 2.8. Anti-Bacterial Experiments

After membrane fabrication, a serial dilution was performed to determine silver’s effectiveness in preventing microbial growth and to determine the microbial concentration that limits silver’s capacity to eradicate bacterial growth over time. A stock solution of *E. coli* BL21 (DE3) was incubated overnight at 37 °C to an optical density reading (ODR) of 0.200 nm, which is approximately equal to 10^−8^ CFU/mL. This allowed us to construct a growth curve for *E. coli*. Following incubation, 7 test tubes containing 9 mL of Luria broth (LB) were set up for a serial dilution of decreasing concentrations ranging from 10^−8^ to 10^−1^ CFU/mL. One ml of the stock solution was removed, placed into the first test tube, and mixed thoroughly. After mixing, 1 mL was removed and placed into the next test tube. This process continued five more times. The functionalized membranes irradiated with UV light were cut to approximately 1 cm^2^ and placed into each test tube. At 1, 2, 4, and 5 h, a 20 µL aliquot was placed on an agar plate, allowed to grow, and evaluated for silver’s capacity to eradicate the growth of *E. coli*. As described by Dumee et al. [18], *E. coli* was grown on an agar plate at 37 °C for 16 h and one colony was transferred to 2 mL of water, high purity VWR. A 100 µL was taken from the solution and streaked onto another agar plate. The PVA-UV membranes in 15 mm diameter were placed on the surface with the PVA-Ag layer facing the agar. The plate was incubated for 24 h at 37 °C.

## 3. Results and Discussion

### 3.1. Membrane Characterization

Figure 1 displays the FTIR spectra of the PVA-Ag membrane. Membranes were analyzed on both sides for desired functional groups and observations of membrane impregnation: Top (the side coated with PVA) and bottom (the side without PVA coating). Four main functional groups were evaluated: Hydroxyl, alkane (C–H stretching), carbonyl, and secondary alcohol (O–C stretching), representing 3280, 2940, 1740, and 1090 cm^−1^, respectively. For the top (coated) surface, the broad hydroxyl peak at 3280 cm^−1^ indicates the presence of PVA on the support membrane. There is also an alkane peak at 2940 cm^−1^. The peak at 1090 cm^−1^ arises from the O–C stretching mode in PVA. For the bottom (uncoated) surface, the hydroxyl and alkane peaks are much reduced, indicating that minimal PVA is present. The peak at 1740 cm^−1^ is created by the carbonyl group in CA. The peak at 1740 cm^−1^ was not clearly visible on the top (coated) surface, which is probably due to the PVA layer obstructing the CA layer from being detected.

SEM micrographs of the composite PVA-Ag membrane are displayed in Figure 2. The cross-section of the PVA-Ag membrane is shown in Figure 2A. The thin PVA layer is visible on top of the support membrane. Due to the commercially purchased support membrane, the CA exhibits a sponge-like sublayer, which is not ideal for ultrafiltration purposes due to hydraulic resistance [31]. Figure 2B shows the top surface of the PVA-Ag membrane. The white particles visible on the membrane’s surface are silver, as confirmed by EDS and XPS, and will be discussed later. The silver is agglomerated on the membrane’s surface and is not evenly distributed.

EDS was performed on the top (coated) surface of the PVA-Ag membrane. Measurements were taken at two locations: on top of a white particle and away from visible white particles (Figure 3). Although silver was detected in both locations, the concentration of silver was higher on a particle (2.37 at %) compared to a location away from visible particles (0.37 at %). This confirms that the silver is somewhat unevenly distributed. Besides Ag, the three other main elements present were carbon (C), nitrogen (N), and oxygen (O). The carbon and oxygen emissions are from the carbon and oxygen present in PVA. The nitrogen emissions may be at least partly from the nitrogen in IDA. The nitrogen concentrations were 3.09 at % on a particle and 2.22 at % at the location away from the particle.

### 3.2. Characterization—Atomic Absorption Content

Studies have shown that immobilizing AgNPs through covalent bond formation leads to slow silver release [18]. Kim et al. immobilized AgNPs into a polyamide (PA) thin-film layer to improve antifouling properties of a TFC membrane [32]. The findings demonstrate that the oxygen binding does stabilize the silver to a certain degree. Atomic absorption spectroscopy was utilized to analyze the bound silver content attached to membranes. When the membranes were irradiated with UV light at 365 nm, a gradual change in color from colorless to a shade of yellow to complete brown was observed with increasing irradiation time. This observation confirms the reduction of Ag^+^ ions. Reducing the valence state of Ag^+^ by photo-reduction allowed membranes irradiated with UV light to retain more silver (23.51 ± 0.98 mg Ag/g of membrane) compared to membranes just functionalized with silver (12.21 ± 4.12 mg Ag/g of membrane). A report from Omrani et al. [21] states that when surfaces are loaded in silver solution, random attachment occurs, which causes weak binding on the surface; whereas reducing silver firmly attaches silver to the membrane surface. In fact, only a small concentration of silver leached out from the PA-UV membranes. The amount of silver leaching into the permeate was reduced by 98% from 10.80 mg Ag^+^/g of the membrane to 0.207 mg Ag^+^/g. Therefore, the immobilized silver ions were either physically adsorbed to the membrane’s surface or weakly chelated to IDA, which allowed the silver to leach out into the permeate. For reduced silver (Ag^0^), the process of dissolution is complex, and the rate depends on a variety of factors such as particle size and the acidity of the environment. Dissolution occurs as the silver is oxidized, then the silver oxide reacts with H^+^ ions to dissolve in water. Therefore, limited release is observed with Ag^0^ [33,34]. These data further support our hypothesis that irradiated (reduced) silver on PVA-UV is significantly more stable than ionic silver. These findings also validate the claim that covalent binding of silver to the hydroxyl groups of PVA prevents leaching, an outcome similar to what was observed with Poly (acrylonitrile-comaleicacid) (PANCMA) [35].

### 3.3. Characterization—X-ray Photoelectron Spectroscopy

XPS multiplex scans of silver and oxygen were performed (Figure 4, Figure 5 and Figure 6) to further investigate any oxidation state changes. Appendix A shows the survey scan of PVA-Ag without UV and with UV membranes. The silver particles presented peaks at Ag3d5/2. C1s and O1s peaks were from the PVA surface layer. A closer look into the Ag 3d5/2 binding energies distinguishes the metallic silver from its oxides. Although published work shows at least three different silver chemical states within the range of 1.2 eV [36,37,38], the binding energies for PVA-Ag and PVA-UV in Figure 4 are 368.60 and 368.23 eV, respectively. The 0.37 eV shift makes it harder to interpret the different oxidation states of silver; however, using the auger parameter, the following values were used to determine the oxidation state of the samples: The Ag 3d spin orbit (doublet) for PVA-A is Ag(3d5/2) 368.60 eV and Ag(3d3/2) 374.62 eV and for the PVA-UV it is Ag(3d5/2) 368.23 eV and Ag(3d3/2) 374.25 eV. Both samples have a doublet split difference of 6.0 eV. This implies that metallic silver (Ag-Ag) is present [39].

Table 1 shows the binding energy values and the calculated Auger parameter. The results of AP suggest that the silver species exhibit different valence states in the membrane structures. The PVA-Ag with silver exists as silver ion, whereas PVA-UV exists as silver metal using the auger parameter calculation [40].

The XPS spectra of O (1s) peaks from PVA-Ag and PVA-UV in Figure 5A,B show binding energy peaks at 532.60 and 532.35 eV, respectively. These binding energies represent silver nanoparticle formation as silver oxide (Ag_2_O), identified at the initial stage of the oxidation [15]. The results demonstrate that the Ag^+^ ions have sp hybrid orbitals, which accept lone pairs of electrons from a hydroxyl oxygen atom. Thus, the silver ion forms a complex with PVA via a linear coordinate bond [15].

The results indicate that Ag nanoparticles have formed. The XPS revealed Ag3d peaks at 368.60 and 374.62 eV for PVA-Ag and Ag 3d peaks at 368.23 and 374.25 for PVA-UV. These data support the claim that Ag^+^ forms oxides (Ag_2_O) identified previously by the O (1s) peak near 532 eV [15]. When photo-irradiated, the Ag^+^/PVA complex lasts for an extended period of time (10 to 30 min). During this time, the hydrated electrons from PVA’s hydroxyl groups act as reducing agents to reduce Ag^+^ to Ag^0^, which allows continued growth of the AgNP on the PVA chain. This claim is supported by results from the XPS overlay spectrum, Figure 4. The intensity of the PVA-UV peak is about 65% higher than the PVA-Ag peak. Previous studies report that the UV-photo-irradiation of Ag^+^ salt, itself, does not produce Ag nanoparticles in the absence of PVA; therefore, PVA acts both as a reducing agent and as a template for the growth of Ag nanoparticles.

XPS analysis was also utilized to examine the binding energies of the silver particles PVA-Ag and PVA-UV as displayed in Figure 6. As a result of the emission of the 3D orbital, the XPS spectra of silver atoms does not yield a single photoemission peak, but a small spaced doublet caused by the spin-orbit splitting of the d-orbitals [22]. The interval between the two peaks is fixed. Thus, only the binding energies of the right peak will be discussed. XPS spectra were taken before and after filtration for both PVA-Ag and PVA-UV membranes. Figure 6A displays the XPS spectra of PVA-Ag. The binding energy of the electrons from the ionic silver on the membrane’s surface was detected at 367.72 eV, and after filtration, XPS analysis indicates that silver was nearly depleted from the membrane’s surface. This suggests that without the UV activation and subsequent silver-PVA complex formation, the membrane would lose its antimicrobial properties and would be more susceptible to biofouling. Figure 6B displays the XPS spectra of Ag(3d) from PVA-UV. Before filtration, the binding energy has a value of 368.05 eV, which was also reported by Huang et al. [5] following the reduction of Ag^+^ to Ag^0^ using dopamine (DOPA). According to the Perkin-Elmer handbook for the Physical Electronics model PHI 5600ci ESCA System [41], this peak remains within the range of reduced silver, and after filtration, the silver peak remains as opposed to the loss of this peak from PVA-Ag. Since the post-filtration peak on PVA-UV is detected at 367.65 eV, which is at a lower oxidation level compared to the pre-filtration peak on PVA-Ag (367.72 eV), the results indicate that silver has been retained on the PVA-UV membrane after pure water filtration. Additionally, the intensity of silver’s photoemission peak on PVA-UV supersedes PVA-Ag, which suggests that the silver concentration on the surface is higher. In addition to utilizing XPS to detect silver’s reduction, the membrane’s discoloration from pristine white to a brownish color was observed. Huang et al. [5] reported that the immobilized silver darkens membranes as a function of its concentration; a phenomenon that we observed as well.

### 3.4. Equilibrium Solution Content

Hydrogel membranes, including pristine unfunctionalized PVA hydrogel control, PVA hydrogel attached to IDA, and functionalized PVA hydrogel attached to silver, were examined for equilibrium solution content (ESC). In addition, both PVA-2.5 and PVA-5 were analyzed for swelling content in water.

PVA-2.5 exhibits an ESC of 85.77%, 88.14%, and 86.25% for unfunctionalized, PVA-IDA, and functionalized, respectively, as compared to PAH-5, which presented an ESC of 78.31%, 80.62%, and 80.2% as seen in Figure 7. As previously stated, the reaction between PVA and GA occurs between PVA’s hydroxyl group and the aldehyde of GA thus forming an acetal bridge [28]. Increasing the concentration of GA in the casting solution increases the amount of cross-linking between the hydroxyl group and aldehyde, which decreases the mesh size of the hydrogel. Moreover, increasing the number of cross-linked PVA chains also decreases the number of water molecules that can be retained. Eventually, this leads to less swelling within the pores of the hydrogel membrane.

We also found that the charged sites within IDA of control membranes are more hydrated compared to either the unfunctionalized membranes or to the IDA-Ag^+^ sites in functionalized membranes where the charge disperses over the metal complex. This result was also observed by Nave et al. [30] who performed a study on PVA hydrogels attached with an IDA-Cu^2+^ complex. It is hypothesized that the PVA-IDA complex expands the pores of membranes, which increases retention of water molecules and promotes swelling in the PVA-IDA complex. In contrast, Ag^+^ substitutes for water molecules and subsequently interacts with IDA’s electron donors. This response lowers water content in functionalized membranes for both PVA-2.5 and PVA-5 hydrogel membranes as seen in Figure 7. The calculated pore size selectively layer, PVA, from the ESC results was an average of 0.018 and 0.015 microns for 2.5% and 5% GA, respectively. The pore size falls in the ultrafiltration range of 0.01–0.1 microns.

### 3.5. Membrane Performance

When membranes fabricated with the hybrid method were analyzed, they all exhibited similar pure water flux measurements. The PWF of PVA-5 (2.61 ± 0.39 L/m^2^·h) displayed an approximate 5% decrease as compared to PVA-2.5′s PWF measurement (2.74 ± 0.74 L/m^2^·h). Increasing the concentration of GA increases the rate of reaction, which leads to a decline in chain mobility and reactivity [27,28]. This increase in GA concentration, in turn, increases the density of cross-links in the PVA structure resulting in smaller pores on the surface as confirmed by ESC. Decreasing the pore size reduces the rate at which water molecules penetrate the cellular matrix, which leads to a reduction in the pure water flux observed.

As previously stated, increasing the concentration of GA from 2.5% to 5% led to an increase in the thickness of the membrane. As a result of this increase in thickness, membranes began to fracture when PVA-5 membranes were incubated in IDA/sodium bicarbonate solution at 60 °C. This result rendered further investigation of silver attachment to membranes fabricated with 5% GA to be impractical, and only membranes fabricated with 2.5% GA were analyzed. The pure water flux of unfunctionalized, functionalized with no UV, and functionalized UV-irradiated membranes are shown in Figure 8. Flux measurements declined, upon the continuous modification of membranes. Pure water flux measurements of PVA-Ag were 2.12 ± 0.98 L/m^2^·h, a 23% decline compared to the pristine PVA-2.5. The PWF of PVA-UV was comparable to PVA-Ag at 2.09 ± 0.21 L/m^2^·h, a 24% decrease from PVA-2.5. Results showing that the attachment of silver onto the membrane yields a lower pure water flux were also reported by investigators who performed an extensive study on polysulfone ultrafiltration membranes [42,43]. As explained by Sawada et al. and Koseoglu-Imer et al., the decline in PWF can be attributed to the barrier effect promoted by the presence of silver particles on the membrane surface and pores [44,45]. The presence of silver on our membranes was confirmed by the SEM micrographs, Figure 2, and EDS, Figure 3.

Filtration experiments using a BSA aqueous solution were used to evaluate protein rejection and fouling of membranes. Ultrafiltration membranes are typically evaluated by their molecular weight cut-off (MWCO), normally defined as a solutes molecular weight that has a rejection coefficient of 80% [46]. Most ultrafiltration membranes have a nominal MWCO ranging from 30 to 1000 kDa [47]. BSA has a molecular weight of 69 kDa, which is within the MWCO range of ultrafiltration, and ideally should have a rejection over 80%. The permeation fluxes of pristine PVA-2.5 and PVA-5 exposed to BSA are opposite to the permeation fluxes obtained using pure water. As GA concentration increased from 2.5% to 5% in the casting solution, the flux increased from 1.87 ± 0.28 L/m^2^·h to 2.09 ± 0.13 L/m^2^·h, respectively, an overall increase of 12%. The flux increase is mainly attributed to the decrease in membrane pore size, which reduces the adsorption or deposition of protein on the membrane surface and within the inner pores of the cellular matrix. In addition to an increase in flux from PVA-2.5 to PVA-5, an increase in rejection was also observed. The rejection of PVA-2.5 was 88% while the rejection of PVA-5 increased to 92%. Therefore, PVA-5 is more suitable for protein filtration tests as compared to PVA-2.5 due to selectivity.

Figure 9 confirms that the decreased pore size results in high flux resistance combined with high filtration flux measurements of PVA-2.5, PVA-Ag, and PVA-UV membranes, underscoring that BSA flux increases with the increased modification of membranes. After the initial addition of silver particles, flux with BSA increased by 8% to 2.02 ± 0.26 L/m^2^·h when compared to PA-2.5, whereas protein permeation flux had a 17% increase to 2.18 ± 0.26 L/m^2^·h following UV irradiation. This increase in flux can be accredited to the improved membrane surface hydrophilicity resulting from the barrier produced by coalesced silver. This barrier prevents BSA agglomeration on the membrane’s surface. In other words, without modification, protein adsorbs to the membrane’s surface due to its hydrophobicity, however, by increasing a membrane’s hydrophilicity following the addition of silver, adsorption of protein onto the membrane’s surface is inhibited. Membrane rejection increased with the increased modification of PVA-2.5 as follows: 88%, 98%, and 99% for PVA-2.5, PVA-Ag, and PVA-UV, respectively (Figure 9). This increase in rejection is also directly correlated to increased flux, increased membrane hydrophilicity, and the barrier effect caused by coalesced silver on the membrane’s surface.

### 3.6. Anti-Bacterial Analysis

A fouling experiment was conducted to test the effectiveness of PVA-UV in inhibiting bacterial growth by observing two variables: Concentration and length of time post-exposure. Our results show that silver effectively inhibits the growth of *E. coli*
Appendix A. After 4 and 5 h of exposure, the agar plates displayed growth for the aliquot representing the 10^−5^ CFU/mL. Bacteria are known to grow at an exponential rate, first experiencing a lag phase, followed by a stationary phase, and ending with a death phase. Unlike the eradication of growth for the 10^−5^ CFU/mL aliquot at 2 h, it appears that after 4 h the concentration of *E. coli* present was able to overcome the membrane’s antimicrobial properties and continue its growth trajectory. In contrast, growth from the 10^−6^ CFU/mL and 10^−7^ CFU/mL aliquots remained suppressed for up to 5 h. The incubation of the membrane with actively growing *E. coli* demonstrates its ability to slow down the growth of bacteria (bacteriostatic effect) over the time period. Growth was observed in dilutions up to 10^−5^ in a 5-h period but the membrane was able to kill the bacteria at the 10^−6^ and 10^−7^ dilutions as demonstrated by no growth after a 5-h period following bacterial exposure. Based on these observations, the silver-impregnated membrane lysed the bacteria present in the 10^−6^ and 10^−7^ dilutions as no growth was observed after the 1 to 5-h growth period had ended. Additionally, the presence of silver particles was found to yield antimicrobial properties as demonstrated after the PVA-UV membranes were exposed to *E. coli* growing on agar plates [18]. The detection of silver is associated with the inhibition of bacterial growth Appendix A. The inhibition is greater around the PVA-Ag-UV membrane than the membrane without Ag. Similar results were demonstrated by others [18,32].

## 4. Conclusions

In this study, we evaluated the membrane performance and biofouling characteristics of membranes fabricated using the hybrid method in combination with silver nanoparticle technology. Our data support the claim that PVA forced within the pores of a CA support membrane using hybrid coating results in higher absorption of CA on the uncoated side, as compared to the coated side. In addition, we tested the hypothesis that reducing silver from Ag^+^ to Ag^0^ enables metallic silver to form nanoparticles, which make it more stable within the membrane’s polymeric matrix and reduces its capacity to enter the permeate during filtration. The data obtained from our metal analysis confirm that membranes exposed to UV reducing conditions retain more silver compared to membranes, which were not reduced. Moreover, our study showed that IDA did not contribute to the stability of Ag attachment, which confirms findings reported by others [48].

In summary, our findings show that a decrease in metal leaching as well as an increase in the stability and duration of a membrane’s antimicrobial activity can be accomplished using a combination of the hybrid coating method and functionalizing the membranes using UV irradiation. UV exposure is particularly important for reducing silver and for promoting the formation of silver nanoparticles. Both of these results contribute to improving the effectiveness of this fabricated membrane, which can be used to provide a more rapid and effective water filtration method. In addition, this new approach to membrane fabrication may also lead to cost reductions due to the extended duration of membrane use.

## Figures and Tables

**Figure 1 polymers-12-01937-f001:**
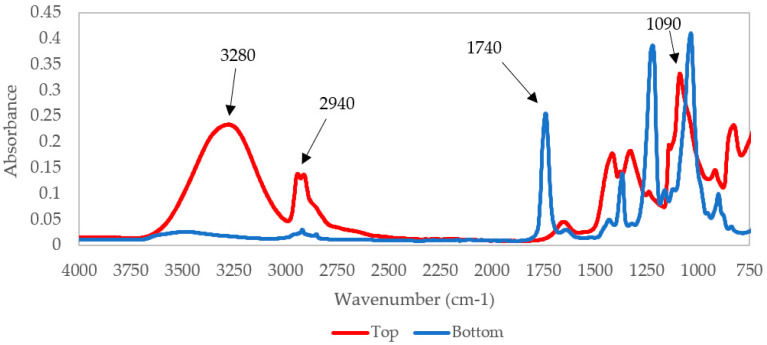
FTIR spectra of the top (coated) and bottom (uncoated) sides of the poly vinyl alcohol (PVA)-Ag membrane.

**Figure 2 polymers-12-01937-f002:**
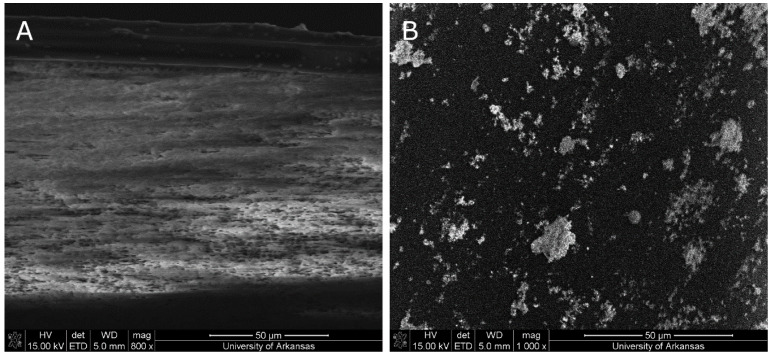
SEM micrograph of (**A**) PVA-Ag cross-section; (**B**) PVA-Ag top (coated) surface.

**Figure 3 polymers-12-01937-f003:**
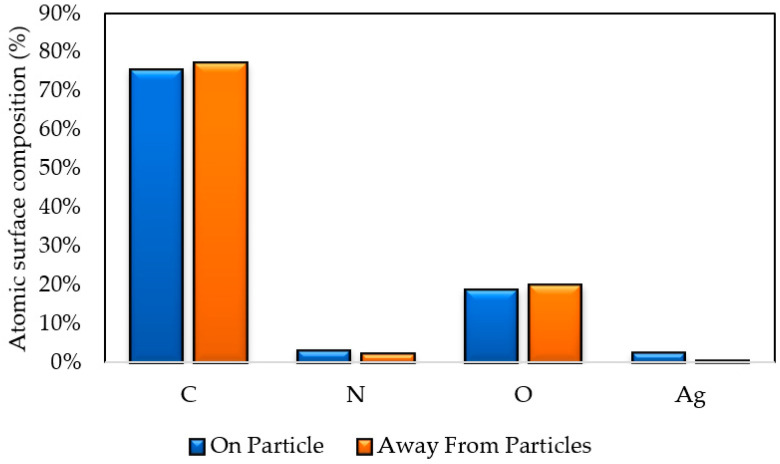
Atomic composition (at %) of the top (coated) surface of a PVA-Ag membrane as determined by EDS.

**Figure 4 polymers-12-01937-f004:**
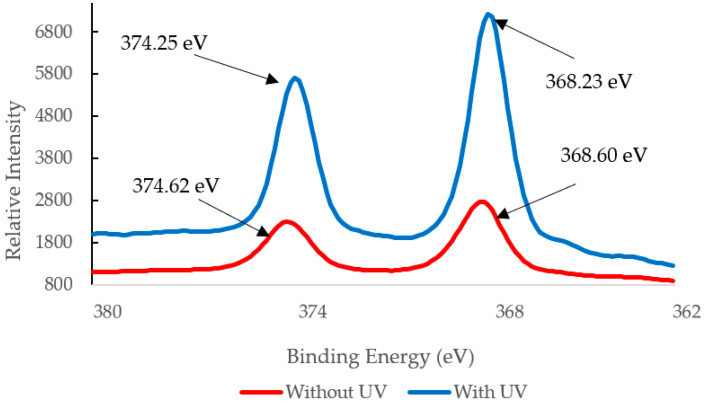
XPS silver 3D region overlay of membrane exposure and non-exposure to UV.

**Figure 5 polymers-12-01937-f005:**
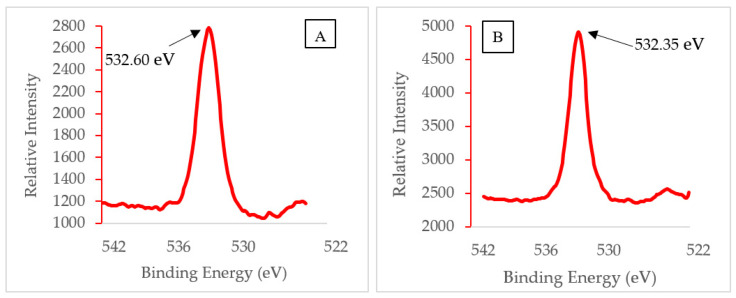
XPS oxygen (1s) spectrum of membranes (**A**) without UV exposure and (**B**) with UV exposure.

**Figure 6 polymers-12-01937-f006:**
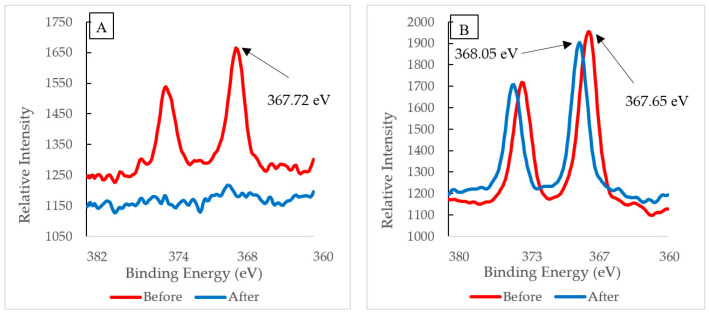
Before and after filtration XPS Spectra of (**A**) PVA-Ag and (**B**) PVA-UV.

**Figure 7 polymers-12-01937-f007:**
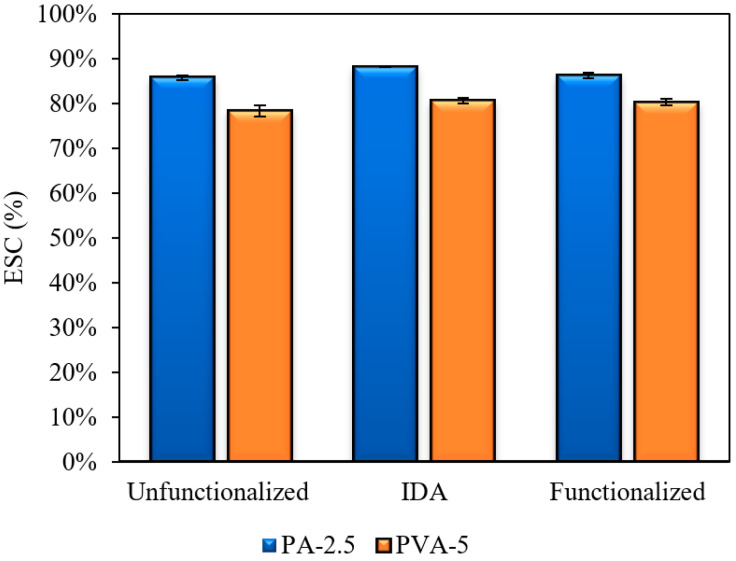
Equilibrium Solution Content (ESC) of control and affinity membranes in water at room temperature.

**Figure 8 polymers-12-01937-f008:**
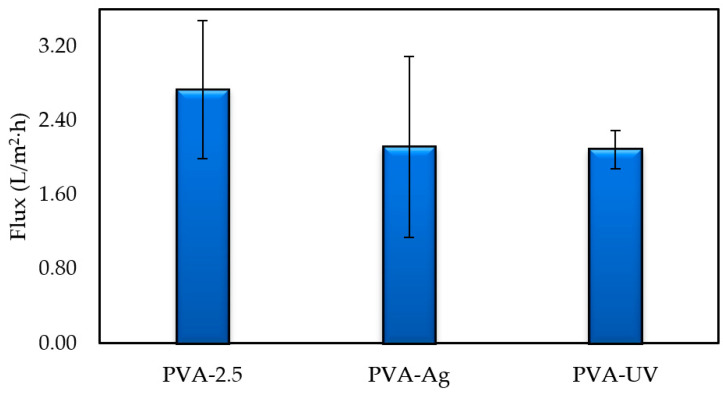
PWF of unfunctionalized and functionalized membranes.

**Figure 9 polymers-12-01937-f009:**
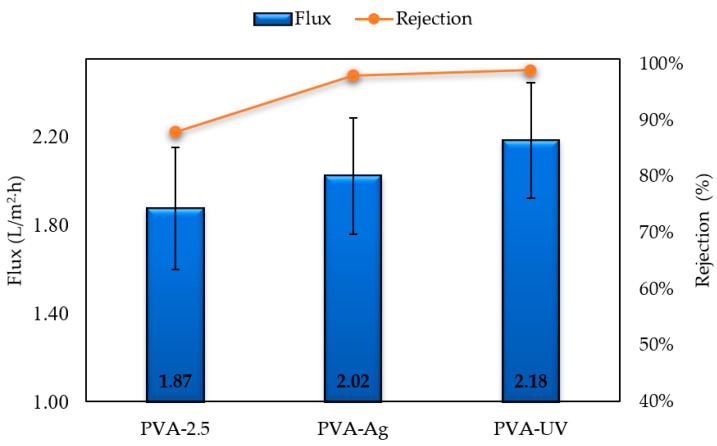
Flux measurements and bovine serum albumin (BSA) rejection of membranes crosslinked with 2.5% GA.

**Table 1 polymers-12-01937-t001:** XPS characteristic peak and auger parameter.

	Ag3d5/2 (eV)	Ag3d3/2 (eV)	KE (eV)	AP (eV)
PVA-Ag	368.60	374.62	355.71	724.10
PVA-UV	368.23	374.25	354.19	722.38

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
