# Peer review of "Development and Characterization of Membranes with PVA Containing Silver Particles: A Study of the Addition and Stability"

_polymers, 2020, doi:10.3390/polym12091937_

Round 1

Reviewer 1 Report

       This paper named “Silver Particles Embedded in Polymeric Ultrafiltration Membranes for Biofouling Resistance” describes the fabrication of UF membrane with PVA and CA filter and then measured the antifouling of this membrane. However, this manuscript is not well prepared. There are some errors in this version such as the wrong subsection order and missing subsection number (3.1 and 3.2). What’s more important, the novelty of this paper is not well addressed in the abstract and introduction parts. We all know that silver is helpful in prevent membrane from biofouling and many researchers have done a lot of work similar with this one. What’s the different between this work and others in literature? Besides, the authors should organize their work in a more logic way and try be keep it clear.

Here are some specific comments.

  • The abstract is too general and I didn’t see the novelty of this work. Many researchers have coated the membrane with Ag to protect the membrane from biofouling. So what’s the main contribution of your work to the available achievements that other researchers have done? This is the key point you should describe in the abstract. Please refine it.
  • The stability of this UF membrane is not shown which is actually very important for the operation of membrane.
  • The biofouling tests is not actual “biofouling” tests. Authors actually measured the survival of bacteria after contacting with the membrane but not the biofouling during filtration. The biofouling should be determined with filtration tests.
  • How about the pore size of the fabricated UF membrane? How did the authors measure it?
  • The authors seem like to use long sentences which makes the readers feel difficult to read. Try to use simple descriptions.
  • Use the full name when it first appears in the manuscript.
  • Some figures are not clear enough. Please replace it with the one with high resolution. For example, Fig. 8.
  • In the figure caption, the authors capitalized the first letter of some words. Please fix it.
  • Too many figure and authors should combine some of them.
  • There are some mistakes in the manuscript. Such as in section 2, it’s should be “2.9 Biofouling experiments” but the authors wrote 3.1. Please check the whole manuscript carefully.

Reviewer 2 Report

The manuscript by Enyinnia et al. presents the design of silver containing membranes for anti-biofouling purpose.

The work is of quality and may be considered after major amendments.

1) The benchmarking of the review and of the discussion is very limited. it is important that that authors compare their works against that already published for silver-containing membranes. Previous works have looked at various incorporation techniques, loadings and encapsulation routes to slow down silver release for instance. This is not well presented and the following publications should be considered and discussed. Others may also be relevant to add.

https://www.sciencedirect.com/science/article/abs/pii/S0376738814008229

https://www.sciencedirect.com/science/article/abs/pii/S0376738816318506

2) The number of figures and their formatting is not always consistent and justified while others are not truly needed in the main manuscript. Figure 1 could go to sup mat. Figure 2 is a copy/paste from the IR software (that is not appropriate at all). Table 1 is not-needed and details could be provided in the text only (which they mostly are anyway). Scale bars are impossible to read on Figure 3. Figure 4 (EDS) is also a poor copy/paste and should be replotted. Figure 5 is not needed as a figure and Table 2 is also not needed the data can be provided in the text only. Figure 6 and 7 could be combined also fitings should be provided for all XPS data since it is otherwise difficult to ensure that the data are useful. Full spectrum should also be provided. Figure 8 is very unclear - not sure if this comes from the authors work? Figure 9 - same comment as 6 and 7. In addition there are weird edges/axis formatting there. It looks like a copy paste too. Figures 12/13 and 14/15 should also be combined. The labels of Figure 14 and 15 are unclear also. It states BSA flux when clearly it should water flux and BSA rejection? Figure 16 is very unclear.

3) Units should be properly used and labelled (hours = h not hrs for instance...). Please proof read the manuscript for consistency. Valid for all units please. The detials provided in the method section are limited and often not sufficient. THis must also be amended.

4) The biofouling experiments were not performed on membranes during filtration. This must be done to substantiate the claims in the title and paper. Without such tests the manuscript would have very limited novelty.

5) A mechanism for the silver release should be proposed and compared to previous works in the area.

Reviewer 3 Report

  1. In the manufacture of a hybrid membrane, a symmetric CA membrane with a pore diameter of 45 μm is impregnated with a PVA + GA solution. In this case, the pore diameter of the supporting membrane should decrease. How large is this decrease and how does the filtration rate decrease? Such information is absent in the article.
  2. A small editing of the references list is necessary:

[3] Zhang, Z., Wu, Y., Wang, Z., Zou, X., Zhao, Y., Sun, L. Fabrication of silver nanoparticles embedded into polyvinyl alcohol (Ag/PVA) composite nanofibrous films through 600 electrospinning for antibacterial and surface-enhanced Raman scattering (SERS) activities. Materials Science and Engineering: C 2016, 69, 462-469, doi:DOI: 10.1016/j.msec.2016.07.015.

[11] Kotel'nikova, N.; Wegener, G.; Paakkari, T.; Serimaa, R.; Demidov, V.; Serebriakov, A.; Shchukarev, A.; Gribanov, A. Silver Intercalation into Cellulose Matrix. An X-Ray Scattering, Solid-State C NMR, IR, X-Ray Photoelectron and Raman Study. Russiab Journal of General Chemistry 2003, 73, 418-426.

[31] VK., K. XPS core level spectra and Auger parameters for some silver compounds. ,. Journal of Electron Spectroscopy and Related Phenomena 1991, 56, 273-277.

[32] Houflund, G.B., Hazos, Z.F., Salaita, G.N. . Surface characterization study of Ag, AgO, and Ag2O using x-ray photoelectron spectroscopy and electron energy-loss spectroscopy. Phys. 683 Rev. B 200, 62, 11126-11133.

[33] Bao, X., Muhler, M., Schedel-Niedrig, Th., Schlogl, R. . Interaction of oxygen with silver at high temperature and atmospheric pressure: A spectroscopic and structural analysis of a strongly bound surface species. Phys. Rev. B 1996, 54.

[38] Son, W.; Youk, J.; Park, W. Antimicrobial cellulose acetate nanofibers containins silver nanoparticles. Carbohydrate Polymers 2006, 65, 430-434.

Round 2

Reviewer 1 Report

I have the following questions or comments regarding your paper:

  1. Modification of membrane using Ag is not a new thing. Please state the novelty of your work in your abstract compared to other work in the literature.
  2. Use the full name when a noun first appears in the article. For example, CA, PVA, XPS.
  3. Please pay attention to the standard expression, for example, coli should be in italic.
  4. The “abstract” should be re-written. This version is not good. An “abstract” should describe the problem in your field in one or two sentences and then propose the technique that you employed to solve this problem. And then show the main results and finally end up with the contribution you made to this field. Right now, there is too much background knowledge in your abstract and too few results. The abstract of your paper is not well written. The novelty of your work should be shown here. Please revise it and make sure that readers can get the important contribution you made to this field after reading the abstract.
  5. The literature review is not wide enough to reflect the state of the art. Please refer to: Environmental Science & Technology 54 (13), 7742-7750, Water Research (2020): 115930; Chemical Engineering Journal 379 (2020): 122351.
  6. This paper is well done but the writing should be more concise. This version is too long.

Reviewer 2 Report

Changes appear appropriate.

Author Response

No response was requested.